# Dishevelled-1 regulates global transcriptomic changes and associates with ETS1 transcription factor

Dalia Martinez-Marin[1,2], Monica Sharma[3], Jenna C. van Wunnik[4], Flávia Sardela de Miranda [5,6], Geetha Priya Boligala[5], Ella C. Jull[7], Grace C. Stroman [7], Rachel L. Babcock [5] & Kevin Pruitt[1,7] ✉

Dishevelled (DVL) is a crucial component of the Wnt-signaling pathway and is vital for multiple physiological processes. Previously thought to have a classically cytoplasmic role, the discovery of DVL nuclear translocation reframed how it is viewed functionally. Although significant progress has been made in understanding the nuclear functions of DVL, further research is required to clarify its roles in transcriptional and epigenetic regulation. A key unresolved question is whether nuclear DVL1 associates with a transcription factor partner. We show here that modulation of DVL1 expression globally affects the transcriptomic landscape. Additionally, analysis of DVL1 ChIP-sequencing allowed us to map genome-wide binding sites, revealing the extensive reach of DVL1 binding. Integration of RNA-sequencing and ChIP-sequencing further revealed ETS1 as a transcription factor binding partner which targets nuclear DVL1 to specific genomic loci. These findings provide insight into the contribution of DVL1 in transcription and clarify aspects of its elusive nuclear function.

Dishevelled (DVL) plays an essential but cryptic role in β-catenin-dependent (canonical) and β-catenin-independent (non-canonical) Wnt signaling. DVLs are the major scaffold proteins in the Wnt pathway and help maintain constitutive oncogenic signaling. DVL proteins govern several cellular processes, including cell proliferation, migration, differentiation, cell polarity and stem cell renewal[1–3]. As a testament to the complex role of DVL, its dysfunctional regulation and/or mutation leads to a range of pathologies associated with Robinow syndrome[4–6], immune dysfunction[7,8], and neurological disorders[9,10]. In addition, emerging evidence suggests DVL is a mediator of cancer malignancy[11–13].

Wnt signaling regulates crucial aspects of development such as cell fate determination, cell migration, neural patterning, and organogenesis[14–16]. DVL serves as a critical hub for transmitting cues from multiple WNT ligands, Frizzled receptors and co-receptors[17,18]. Historically, DVL's cytosolic role has been studied almost exclusively. These studies uncovered its role in protein scaffolding, β-catenin stabilization, and signalosome formation[19–21]. However, recent studies show DVL nuclear localization/export signals (NLS/NES)[22,23], transcription factors such as FOXK1 and FOXK2[24], as well as certain critical lysine residues[23,25] play a key role in regulating DVL nuclear localization. Nuclear DVL has been shown to bind to Wnt target genes[24,26], promoters that drive aromatase overexpression in cancer, and play a cell-type and paralog-dependent role in activating or repressing *CYP19A1* tissue-specific promoters for transcription[27]. In addition, a recent report highlighted the potential for DVL transcriptional

[1]Lineberger Comprehensive Cancer Center, University of North Carolina, Chapel Hill, NC, USA. [2]Center for Nanotechnology in Drug Delivery, University of North Carolina, Chapel Hill, NC, USA. [3]Department of Immunology and Molecular Microbiology, Texas Tech University HSC, Lubbock, TX, USA. [4]Department of Biology, Texas Tech University, Lubbock, TX, USA. [5]Department of Cell Biology and Biochemistry, Texas Tech University HSC, Lubbock, TX, USA. [6]Breast Center of Excellence, Texas Tech University HSC, Lubbock, TX, USA. [7]Department of Pharmacology, University of North Carolina, Chapel Hill, NC, USA. ✉e-mail: kevin_pruitt@med.unc.edu

regulation of immunomodulatory genes[11,28]. Despite our advances in elucidating DVL nuclear role, the scope of DVL involvement in transcription and the transcription factors that interact with nuclear DVL proteins remain a major unknown.

This present report utilizes an integrative analysis approach to uncover an additional dimension of DVL regulation. Collectively, our findings identify a list of candidate transcription factors that potentially bind nuclear DVL1. We show that ETS1 binds nuclear DVL1 and plays a key role in targeting it to gene promoters in which they co-localize. This provides insight into the contribution of DVL1 in the transcription of genes involved in various cellular functions, including those that drive triple-negative breast tumorigenesis and modulation of the tumor microenvironment.

## Results

### Increase in DVL1 expression results in global changes in gene expression

Previous publications have shown increased DVL1 in breast cancer tissue adversely associated with clinical risk factors[28,29], yet the impact of DVL1 on global gene expression has yet to be investigated. To assess the role of DVL1 in regulating transcriptomic changes, we performed RNA-seq and identified differentially expressed genes in MDA-MB-231 cells with increased DVL1 (DVL1$^{OE}$) expression relative to empty vector (EV) control.

Increased DVL1 expression led to significant (padj <0.05) differentially expressed genes (DEGs) where more than half of the transcriptome was altered in EV vs DVL1$^{OE}$ (Fig. 1a, *left*). We utilized the statistical method TREAT (t-tests relative to a threshold)[30] to focus specifically on differentially expressed genes (DEGs) exhibiting a minimal fold change of 10%, while simultaneously maintaining a low false discovery rate. This approach enables us to pinpoint mRNAs that exhibit a substantial and statistically significant fold change, serving as an initial step in our analysis. Similar numbers of genes were up-regulated and down-regulated (up-regulated: 5399; down-regulated: 5715). Classification of DEGs revealed that while most genes were protein coding, some surprising changes were uncovered, such as the 8 affected mitochondrial tRNA (Fig. 1a, *right*), suggesting a larger breadth of DVL1 regulation than previously described.

We performed functional pathway analysis, and the DEGs fell into diverse functional categories, including pathways involved in development, immune response, signal transduction, and cell migration and adhesion (Supplementary Fig. 1A, B). This indicates that DVL1 activity affects/regulates multiple cellular processes. In addition, this gene set enrichment analysis (GSEA) was used to identify DEGs enriched in gene sets defined by mining large collections of cancer-oriented microarray datasets, signatures of pathways frequently dysregulated in cancer and hallmark gene sets that represent specific well-defined biological processes (Supplementary Table 1). Further analysis of DEGs using ClinVar[31], a publicly available archive that aggregates genetic data by variant-disease phenotypes, revealed the relationship between DEGs and the possible clinical effect of their altered expression due to manipulation of DVL1 expression (Supplementary Fig. 1C). Together, these data show the importance of DVL1 in regulating the transcriptome.

RT-qPCR results in different cell lines mirror the RNA-seq results, revealing that similar up- or down-regulation of transcripts due to increased DVL1 expression (Fig. 1b). Selected transcripts with significant roles in a range of cellular processes include APCS (serum amyloid P component), MMP1 (high matrix metalloproteinase 1), BMP2 (bone morphogenetic protein 2), CXCR4 (CXC motif chemokine receptor 4), SIRPA (signal regulatory protein α), and TNFRSF18 (tumor necrosis factor receptor superfamily member 18). The presence of near-equal up-regulated and down-regulated DEGs provides evidence that DVL1 may act in a context-dependent manner as either an enhancer or repressor of transcription.

Though previous studies have shown that DVL localizes to genes critical for immune system modulation and metabolic regulation[11,23,28], the role of DVL1 in global transcriptional regulation remains unknown. To address this crucial knowledge gap, we began with a comprehensive analysis of the genomic binding patterns of DVL1. ChIP-seq analysis in MDA-MB-231 of endogenous DVL1 revealed an impressive spectrum of loci to which DVL1 localizes. Investigation of DVL1 peak distribution across chromosomes shows global localization (Supplementary Fig. 2a). Given that DVL1 is a key player in Wnt signaling, we analyzed the distribution of DVL1 binding peaks across genes associated with this pathway. Our findings revealed that DVL1 binds to numerous Wnt-associated genes, highlighting its significant role in the regulation of this critical signaling pathway (Supplementary Fig. 2B and Supplementary Table 2). To further extend our ChIP-seq analysis, we focused on DVL1 binding at exonic and non-exonic regions in relation to transcription and epigenetic markers. At specific exonic regions, DVL1 binding overlapped with histone markers commonly associated with active enhancers (H3K27ac, H3K4me1), active promoters (H3K4me2, H3K4me3, H3K9ac), and other markers of gene transcription (H3K36me3, POL2RA) in genes linked to tumor progression and metastasis[32–34], hormone dysregulation linked with tumorigenesis[35] and developmental disorders[36] and Wnt signaling[37]. DVL1 binding at non-exonic areas, including introns and intergenic regions, highlights the extensive global binding capacity of DVL1 (Fig. 1c).

Given the long-standing question concerning the role of nuclear DVL, we next wanted to determine the relative frequency of the types of genomic regions to which DVL1 localizes. Feature distribution shows about 50% of all DVL1 peaks are in the promoter region of the gene hits (Supplementary Fig. 3B, *Top*). In addition, the UpSet plot shows the majority of DVL1 peaks are in the genic, intron, exon, promoter, and 3'UTR regions (Supplementary Fig. 3B, Lower). A unique set of peaks in the intergenic regions suggests another function for DVL1 regulation of nearby genes, apart from promoter activity. In addition, we find DVL1-peak enrichment and genome-wide correlation with putative cis-regulatory modules (CRMs), such as promoters (H3K4me3 and H3K27me3), enhancers (H3K27ac and H3K4me1), and repressors (H3K79me3) (Supplementary Fig. 3C).

As there was significant enrichment near the transcription start site (Supplementary Fig. 3A), and since DVL1 does not have a canonical DNA binding domain, we reasoned that DVL1 may be associated with genomic loci via interaction with a transcription factor. Consequently, we searched for transcription factor consensus sequences within the DVL1 ChIP-seq reads and utilized known sequences of core promoter elements to identify the genes in which DVL1 binds to those elements. ChIP-seq analysis identified 9438 DVL1-peaks matching the core-promoter sequences across the human genome, which can be assigned to 3122 genes. We sought to detect enriched DNA sequences shared among the identified DVL1-peaks by performing a de novo motif search. We identified core-promoter element motifs for the TATA-box, CAAT-box, and Inr regions (Supplementary Fig. 3D, *Top*), as well as a known member of the β-catenin/TCF transcriptional complex, TCF4 (Supplementary Fig. 3E). To further validate our in-silico analysis, we selected genes with significant peak enrichment to represent each of the identified core-promoter element motifs and performed ChIP-qPCR using endogenous DVL1 antibody (Supplementary Fig. 3D, *Bottom*).

To identify which genes DVL1 is directly regulating, we combined the RNA-seq and ChIP-seq profiles and identified 633 genes that are differentially expressed (padj < 0.005) and contain significantly enriched DVL1 binding loci (Supplementary Table 3). As in the RNA-seq analysis, there were similar numbers of up- and down-regulated genes (up-regulated: 339; Down: 294). Reactome analysis identified several significantly dysregulated (nominal [NOM] *P*-value < 0.05) pathways, with cell signaling and communication, immune regulation, and

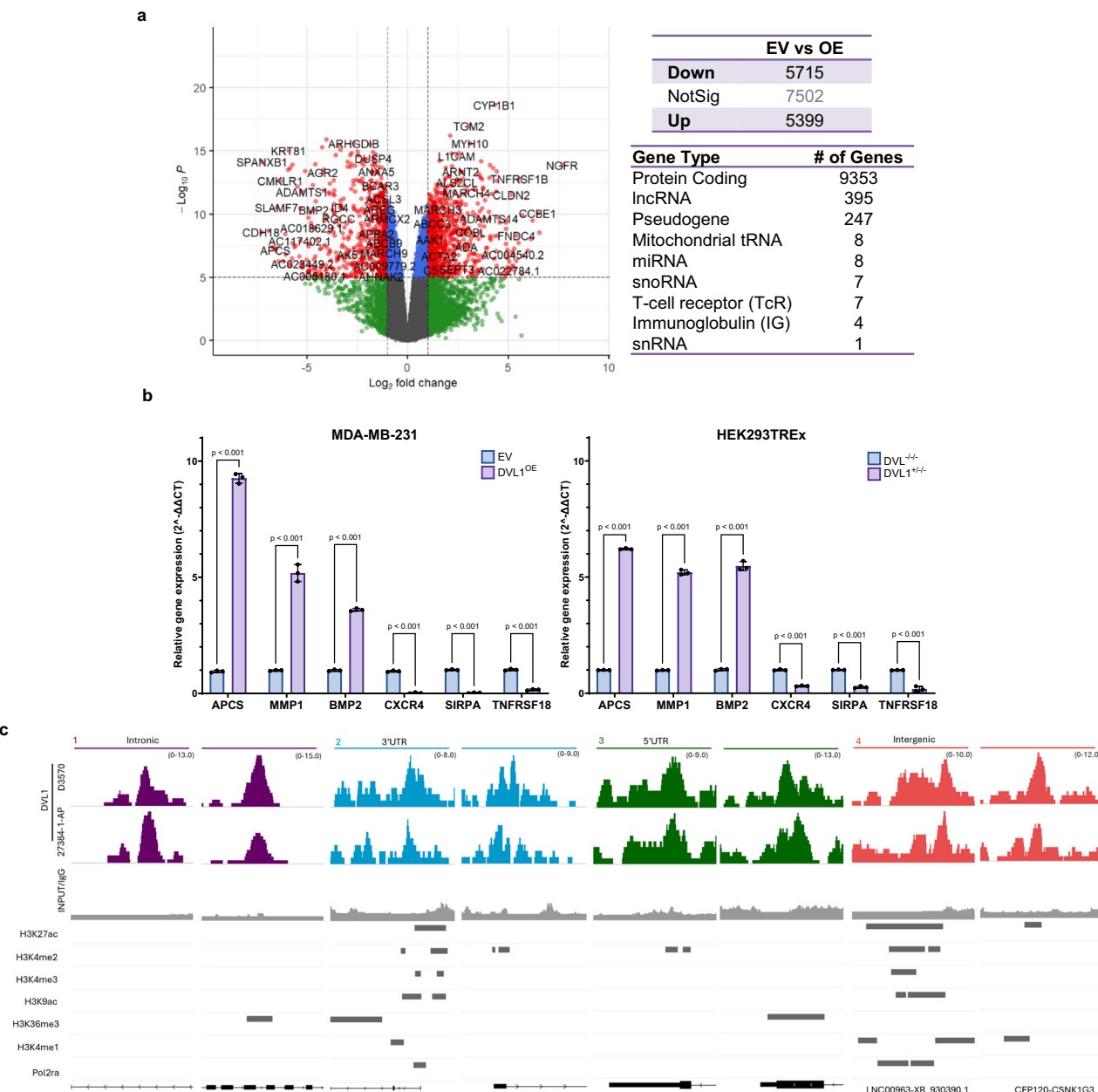

**Fig. 1 | Evaluation of triple negative breast cancer cell line MDA-MB-231 with increased DVL1 expression reveals numerous differentially expressed genes (DEGs) and core promoter binding. a** Differential gene expression analysis of RNA sequencing of MDA-MB-231 with increased expression of DVL1 (DVL1^OE) in comparison with control cells (EV). Number of significantly expressed (n = 3 biological replicates, log2FC > 2 or < − 2, padj < 0.05, statistical analysis and p-value calculations were performed using EdgeR) genes after *TREAT, Top Right*; Categorical placement of genes to identify gene family type, *Bottom Right* (**b**) RT-qPCR (n = 3 biological replicates (3 technical replicates each), mean ± SD, two-way, ANOVA followed by Sidak's multiple comparison test was used to test significance) of *APCS, MMP1, BMP2, CXCR4, SIRPA, TNFRSF18* in MDA-MB-231 control cells (EV) and 231 cells with increased DVL1 expression (DVL1^OE), and HEK293TREx DVL triple knock out (DVL^-/-/-) control cells and HEK293TREx DVL-TKO with increased DVL1 expression (DVL^+/-/-). **c** Visualization of DVL1 (D3570, 27384-1-AP), and negative control genomic region binding peaks in IGV in intronic regions, 3'UTR, 5'UTR and intergenic regions, overlayed with RefSeq, and markers for H3K27ac, H3K4me2, H3K27me3, H3K4me1, H3K9ac, H3K36me3, and POL2RA. Source data are provided as the Source Data file.

signaling cascades being the most enriched (Supplementary Fig. 4). Gene ontology (GO) analysis of biological and molecular functions revealed significant (nominal [NOM] P-value < 0.05) enrichment in multiple cellular processes. The top three being in system development (differentiation, morphogenesis, development), regulation of organismal process (cytokine production, antigen processing and presentation, cellular response to external stimulus, cell-cell communication), and migration (Supplementary Fig. 5A).

To better understand the influence of genes whose expression is influenced based on DVL1 amounts, and the impact of aberrant DVL1

on disease, compartmentalization and ClinVar pathway analysis was performed on our RNA/ChIP-seq gene set. The genes regulated by DVL1 extended across all compartments of the cell (nucleus, nuclear membrane, cytoplasm, extracellular membrane) as well as genes associated with the transcription regulator complex (Supplementary Fig. 5B) and affected pathologies related to both development and cancer (Supplementary Fig. 5C). Taken together, the integrative analysis of RNA-seq and ChIP-seq identified numerous DVL1 regulated genes of different biological functions and their potential phenotypic effects.

## DVL1 co-localizes with Transcription Factors at the Promoter

While the functional significance of nuclear DVL paralogs is beginning to emerge[38], virtually nothing is known about nuclear DVL1 and whether or how it partners with transcription factors. We used locus overlap analysis (LOLA) to test overlap sets of genomic regions in our DVL1 ChIP-seq data and publicly available datasets and motifs using JASPAR TFBSTools, an R/Bioconductor package for the analysis of transcription factor binding sites and their associated transcription factor profile matrices, to identify possible DVL1-transcription factor binding partners at the promoter regions. LOLA utilizes the padj value, odds ratio (from a Fisher's exact test), and the raw number of overlapping regions to create an aggregate score and ranks each pairwise comparison for each of these statistics to calculate a combined rank for each factor (Fig. 2a). Using this approach, we were able to obtain a list of 35 transcription factors (Supplementary Table 4) with significant binding enrichment (p.valueRnk) to the same sites of DVL1 binding enrichment (Supplementary Table 3). Fig 2b, *top*, shows the 10 highest ranked (maxRnk) transcription factors, indicating potential binding partners of DVL1 at genomic loci in the promoter region.

Previous studies[39] and western blot analyses have shown the top predicted value transcription factor, GATA2, has minimal expression in MDA-MB-231 (Fig. 2b, *bottom*), therefore we further investigated the second highest ranked transcription factor ETS1 (ETS proto-oncogene 1, transcription factor). ETS1 regulates important biological processes such as cell proliferation, differentiation, apoptosis and angiogenesis, and contributes to the initiation and progression of diverse tumors, including breast cancer[40]. We assessed ETS1 expression in MDA-MB-231 cells expressing EV or HA-tagged-DVL1 (DVL1$^{OE}$). In addition, we utilized a HEK293TREx cell line that is devoid of all three DVL paralogs (HEK293TREx-TKO; DVL$^{-/-/-}$) and a HEK293TREx triple knockout cell line with an HA-tagged-DVL1 reintroduced (HEK293TREx-TKO-DVL$^{+/+/-}$). Expectedly, ETS1 protein expression was largely in the nuclear fraction, thus allowing us to study DVL1-ETS1 physical interaction in the nucleus (Fig. 2c). Nuclear extracts were used for co-immunoprecipitation of DVL1 (HA) and ETS1 which confirmed our in-silico results and revealed DVL1-ETS1 interaction (Fig. 2d). Several genes (*BMP2, CD96, COL1A1, DDX3X, PTEN, SOX4*) previously identified in our RNA/ChIP-seq integrated analysis were analyzed and were shown to have altered expression in phenotypes identified in our ClinVar analysis (Supplementary Fig. 5C). These targets are also known to be regulated by ETS1.

We assessed the binding potential of ETS1 to the same genomic loci as DVL1 in our two models and found ETS1 binds to the same loci using ChIP-qPCR (Fig. 2e). To further assess DVL1 and ETS1 co-occupancy at this six-gene panel, reciprocal DVL1/ETS1 and ETS1/DVL1 ChIP/re-ChIP-qPCR was performed (Fig. 2e). These results further implicate DVL1 as a transcriptional regulator and identify a transcription factor binding partner. Together, this suggests that DVL1 may have the potential to bind to other transcription factors at specific genomic loci as observed with respect to ETS1.

## ETS1 is Required for DVL1-DNA Association

DVL1 has yet to be shown to have DNA-binding motifs, suggesting its association with genes at the promoter region requires a transcription factor binding partner. To determine whether ETS1 targets DVL1 to specific promoters, and is required for DVL1 promoter occupancy, we used three models of DVL1 gene expression: loss-of-function (DVL1$^{KD}$, DVL$^{-/-/-}$), control (EV, Parental), and gain-of-function (DVL1$^{OE}$, DVL$^{+/+/-}$). We first sought to identify how manipulation of DVL1 expression affected our selected gene panel. Gene expression analysis showed opposing results between low and high DVL1 expression (Fig. 3a), suggesting that DVL1 regulates transcription activation and repression.

Having identified a surrogate panel of genes to which DVL1 binds their promoters and regulates mRNA levels, we next determined the relationship between DVL1 and ETS1 at these loci. To test whether DVL1

is required for ETS1 binding to the genomic loci, we performed ChIP for endogenous DVL1 and ETS1, as well as ChIP/re-ChIP, in the above-described gain/loss of function and rescue models. Our data suggest the loss of DVL1 does not affect ETS1 binding to DNA at these specific loci (Fig. 3b and Supplementary Fig. 6). We next evaluated whether the loss of ETS1 influences expression of the surrogate gene panel and DVL1 localization to their promoters. ETS1 knockdown relative to non-targeting control (NTC) led to significant differential gene expression (Fig. 3c). ETS1 knockdown not only significantly reduces ETS1-DNA binding as expected, but strikingly, also significantly reduces DVL1 localization to the same genomic loci (Fig. 3d). These data reveal that DVL1 requires ETS1 for localization to the gene promoter panel to which DVL1 and ETS1 co-localize.

Our data demonstrate that during basal conditions, DVL1 and ETS1 partner at the same genomic loci to regulate transcription activation and repression in genes such as *BMP2, CD96, COL1A1, DDX3X, PTEN*, and *SOX4* (Fig. 3e, *top*). Repression of DVL1 does not prevent ETS1 binding to the promoter region but does alter the expression of ETS1-regulated genes (Fig. 3e, *bottom left*). This suggests that nuclear DVL1 might act in its conventional role as a scaffolding protein to stabilize transcription by ETS1. In addition, reduction of ETS1 prevents DVL1 from binding, and again alters transcription (Fig. 3e, *bottom right*). Together, these data suggest that DVL1 is dependent on ETS1 for promoter occupancy and that DVL1 regulates transcription activation and repression.

## SIRT1 Regulates ETS1 and DVL1 transcriptional abilities

SIRT1, a class III lysine deacetylase, is known to positively regulate DVL1 protein levels in cancer cells[41], and promote DVL1 scaffolding of proteins, such as TIAM1[3]. In addition, it is known that aberrant lysine acetylation of non-histone proteins can have a large effect on biological processes[42] such as epigenetic reader function[43], estrogen synthesis[44], DVL subcellular localization[23], as well as many cancer-associated processes[45]. Given the known interaction between SIRT1 and DVL1, we evaluated the influence of SIRT1 on DVL1/ETS1 transcriptional regulatory ability. Reciprocal co-immunoprecipitation of endogenous DVL1, ETS1, and SIRT1 demonstrated DVL1/SIRT1 and DVL1/ETS1 interaction (Fig. 4a), but SIRT1/ETS1 interaction was absent. Interestingly, ChIP and ChIP/re-ChIP/re-ChIP demonstrated that all three proteins bind to the same genomic loci (Fig. 4b and Supplementary Fig. 7) of our ETS1-regulated gene panel.

Pharmacological inhibition of SIRT1, using 32 nM SIRT1-specific inhibitor IV (SIRT1 IV$_i$), has been shown to influence acetylation patterns on endogenous DVL1, which alter DVL1 localization and binding to promoter regions[23]. Therefore, we investigated the effect of SIRT1 inhibition on DVL1/ETS1 transcriptional ability. Addition of SIRT1 IV$_i$ showed a significant change in ETS1 regulated genes, not surprisingly having a cell line dependent effect (Fig. 4c). ChIP for DVL1(HA), ETS1, and SIRT1 after treatment with SIRT1 IV$_i$ showed a significant reduction in binding of all three proteins to the genomic loci in both cell lines (Fig. 4d).

These data suggest that under basal conditions, DVL1 binds to both ETS1 and SIRT1, along with other possible cofactors yet to be discovered, in the promoter region to regulate gene expression (Fig. 4e, *top*). In MDA-MB-231 cells, when SIRT1 is inhibited, both DVL1 and ETS1 fail to bind to their genomic loci, dysregulating gene expression (Fig. 4e, *bottom left*). However, in the HEK293TREx cell line, when SIRT1 is inhibited, DVL1 and ETS1 have a significant reduction in binding potential, but it still has a significant effect on gene expression (Fig. 4e, *bottom right)*. Together, these data suggest a role of DVL1 as a transcriptional regulator.

## Discussion

Dishevelled was first recognized in the development of *Drosophila* and has since been shown to play a significant role in a vast and crucial

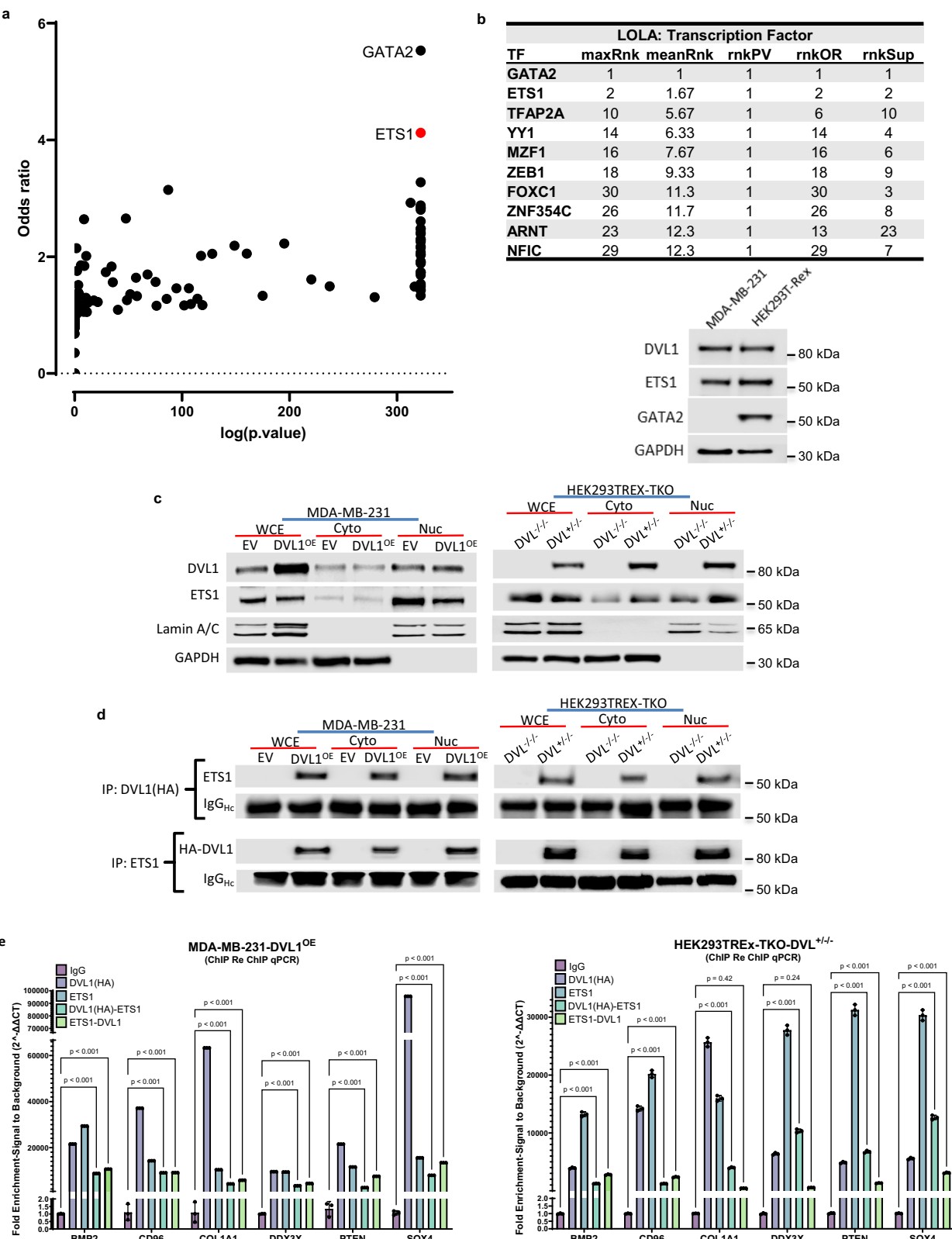

array of functions such as cell proliferation, differentiation, cell polarity, and body axis formation during development[46,47]. While the cytosolic function of some DVL paralogs has been clarified with recent work from several labs[1,48], its nuclear role remains an enigma[38]. Studies have demonstrated that DVL, through its NLS (nuclear localization signal) and NES (nuclear export signal) domains, can shuttle between the cytoplasm and the nucleus[22,23], likely influenced by post-

translational regulation of certain critical lysine residues[23,25]. This dynamic localization of DVL offers potential insights into its dual impact on cellular signaling. It not only suggests significant implications for Wnt-associated gene transcription, indicating that the nuclear presence of DVL might directly influence gene expression patterns, but also raises the possibility that shifts in DVL localization could deplete cytosolic DVL levels, thereby affecting its ability to activate Wnt

**Fig. 2 | DVL1 associates with transcription factor ETS1. a** Locus overlap analysis (LOLA) was utilized to test the overlap sets of genomic regions in our DVL1 ChIP-seq data and transcription factor profile matrices, to identify possible DVL1-transcription factor binding partners at the promoter regions. The LOLA scatter-plot visualizes data on 3 metrics (*p*-value, odds ratio, and raw number of overlapping regions) to assess overlap. Each query-to-database set interaction is a point. The axes measure the *P*-value and odds ratio of each comparison, and points in the upper-right region of the plot are the best hits for all 3 metrics. **b** Top 10 LOLA comparisons of DVL1 ChIPseq (*n* = 3 biological replicates, log2FC > 1 or < − 1, padj < 0.05, peak identification and statistical analysis were performed using MACS2). maxRnk is the aggregate score that shares strengths of each of the individual statistics: rnkPV (*p*-value rank), rnkOR (odds ratio rank), rnkSUP (support rank), *Top*, western blot of top two predicted TFs GATA2 and ETS1 in MDA-MB-231 and HEK239T-Rex, Bottom. Representative blots shown (three biological replicates); GAPDH was the loading control. **c** Western blotting of DVL1 and ETS1 in whole cell extract (WCE), cytoplasmic fraction (cyto), and nuclear fraction (nuc) in MDA-MB-231 and HEK293TREx-DVL-TKO. Representative blots shown (three biological replicates); Presence or absence of GAPDH was the loading control for whole cell extract (present), cytosolic fraction (present), and nuclear fraction (absent). Lamin was used as a secondary loading control for the nuclear extract. **d** Co-Immunoprecipitation for HA-DVL1 and ETS1, and probe for their respective partner in the cytoplasmic and nuclear fractions. Representative blots shown (three biological replicates); IgG heavy chain was the loading control for IP. **e** ChIP and ChIP/re-ChIP (*n* = 3 biological replicates (3 technical replicates each), mean ± SD, two-way, ANOVA followed by Dunnett's multiple comparison test was used to test significance) for HA-DVL1 and ETS1 in the same genomic region in MDA-MB-231 and HEK293TREx cells in genes identified in ChIP-Seq and are known to be regulated by ETS1. Source data are provided as the Source Data file.

signaling. This interplay between DVL's localization and Wnt pathway activation is crucial for understanding the broader regulatory mechanisms in cellular signaling. Though recent reports highlighted the role of DVL3 and DVL1 in the regulation of transcription of various immune cell-related genes[11,28], the scope of DVL transcriptional regulation has yet to be identified. In addition, different DVL paralogs were previously described to bind to promoters that induce aromatase overexpression in cancer and also exhibit a paralog-dependent and cell type-dependent role in activating or repressing *CYP19A1* transcription[27]. DVL3 was shown to interact with c-Jun[26] on the promoter of a Wnt target gene, Forkhead box transcription factors (FOXK1/2) have been shown to play a key role in nuclear translocation of DVL2[49], and DVL1 bound with p65 repressed NFκB to target gene expression[50]. Our ChIP-seq analysis has revealed the presence of motifs within the promoter regions of genes potentially impacting Wnt signaling, including those for TCF. This discovery suggests that DVL1 may be a consistent component of the β-catenin transcriptional complex. However, additional experiments are necessary to validate this role and elucidate the specific interactions and functional implications of DVL1 within this complex[26]. The broad peak coverage of DVL1 at both exon/intron regions could suggest DVL1 interactions with splicing factors. Previous studies have employed ChIP to reveal the recruitment of additional splicing factors and components of the standard spliceosome in co-transcriptional splicing, identifying specific DNA sequences linked to a higher occurrence of factors associated to intron retention events[51,52]. Interestingly, ChIP-seq experiments on transcription factors have identified the assembly of spliceosome components on individual genes and has shown RNA Polymerase II (RNAPII) to associate with splicing factors, with most introns initiating their splicing co-transcriptionally[53–55]. Thus, since splicing components are concentrated at intron/exon junctions and RNA-binding proteins are stabilized on nascent RNA through formaldehyde cross-linking, specific peaks may represent splice junctions and offer insights into DVL1 binding enrichment at certain loci. Further investigation is required to explore the potential interaction between DVL1 and the co-transcriptional splicing machinery. Integration of our ChIP-seq data with several databases and data mining tools identified possible DVL1-transcription factor binding partners at the promoter regions of DVL1-regulated genes. One of the top hits is ETS1 (ETS proto-oncogene 1, transcription factor), and is known to regulate important biological processes such as cell proliferation, differentiation, apoptosis, and angiogenesis[56,57]. In addition, ETS1 has been shown to activate Wnt/β-catenin signaling by regulating TGM2 transcription[58]. Aberrant ETS1 expression has been implicated in the progression of numerous malignancies and promotes tumor metastasis[40,59,60]. However, recent studies have proposed a contrasting role of ETS1 as an anti-oncogene, suggesting a dichotomous role of ETS1 in tumorigenesis[40,61]. The interaction between DVL and ETS1 represents a crucial molecular mechanism the drives transcriptional regulation in cellular processes. This interaction underscores the complexity of signaling pathways

involved and highlights potential targets for therapeutic interventions where these pathways are dysregulated.

In addition, with integrative analysis of DVL1 binding loci and gene expression data, we identified the genes that are bound by DVL1 and are differentially expressed when DVL1 expression is altered. This interaction highlights additional functional possibilities for DVL1, a protein whose primary role was once thought to be only cytosolic. Our findings identify DVL1 as a transcriptional regulator capable of activating or repressing transcription through partnering with transcription factor ETS1 and binding together at the same genomic loci. These results have obvious implications regarding the role of DVL1 in cellular function, including tumorigenesis. Upon overexpression of DVL1, a significant portion of the transcriptome exhibits altered expression levels. While only a minority of these transcripts show DVL1 binding only in the promoter region, according to our ChIP-seq data, a considerably larger number of genes display DVL1 binding at multiple locations along the gene. This observation suggests that DVL1's influence on the transcriptome extends beyond its interaction with promoter-bound transcription factors like ETS1. DVL1 may also impact gene expression by interacting with splicing factors or by influencing RNA stability and degradation processes. This multifaceted role highlights DVL1's complex involvement in gene regulation, suggesting mechanisms that extend across different stages of mRNA processing and stability. While our study does not resolve which transcriptomic changes are associated strictly with DVL1 cytosolic vs. nuclear function, it does assess global transcript changes linked with diverse cellular and organismal functions. Recently, DVL2 was shown to undergo liquid-liquid phase separation (LLPS) to stabilize β-catenin by recruitment of Axin into a biomolecular condensate at the plasma membrane[48,62]. While the propensity of other DVL paralogs to undergo LLPS remains unknown, it is interesting to consider the implications of cytosolic vs. nuclear DVL biomolecular condensates and whether transcription factors are recruited to these dynamic structures. Furthermore, understanding whether these nuclear interactions of DVL involve its monomeric or dimeric forms, or even heteromeric combinations with other DVL paralogs, could provide more detailed insights into the molecular configuration of DVL when it associates with nuclear targets. This knowledge is crucial for unraveling the complexities of DVL function and its role in signal transduction pathways. Moreover, given the critical role of post-translational regulation in oncogenic signaling[43,44], it will be important for future studies to assess the role of DVL PTMs and regulating cell migration/metastasis[41] as well as epigenomic changes[45].

## Methods

### Cell culture
Triple-negative breast cancer cell line MDA-MB-231 was purchased from ATCC. Cells were authenticated with STR technology and used in a low passage (< 20) within 6 months or less after receipt or resuscitation. HEK293TREx DVL triple-knockout (TKO) was a generous gift

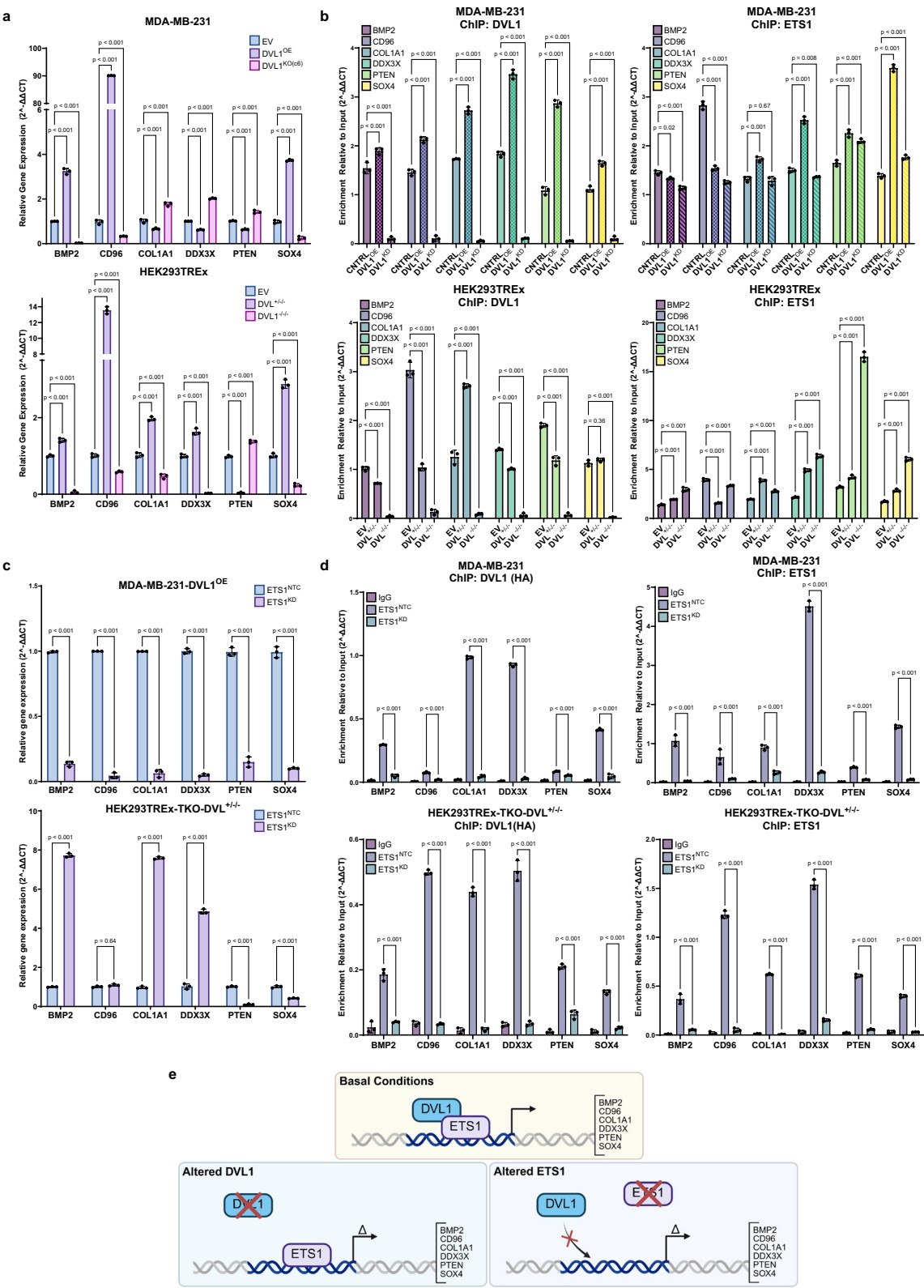

from the lab of Vítězslav Bryja[63]. MDA-MB-231 and HEK293TREx DVL TKO cells were cultured in DMEM (Gibco, 11995-065), supplemented with 10% fetal bovine serum and 1% penicillin/streptomycin (Gibco, 15070-063).

**Chromatin immunoprecipitation (ChIP)**

Cells were grown to confluence to a final count of approximately 35 million cells per plate. Cells were cross-linked with 1% (w/v)

formaldehyde for 8 min and quenched in 0.125 M of glycine for 5 min at room temperature. Media was removed and cells were washed thrice with sterile ice-cold 1X PBS. Cells were scraped in 1X PBS plus protease inhibitor (PI) cocktail (ThermoFisher) and centrifuged to obtain a cell pellet. Pelleted cells were resuspended in SDS Lysis buffer (50 mM Tris-HCl pH 8.0, 10 mM 0.5 M EDTA, and 1% SDS) with a protease inhibitor cocktail and then sonicated in a Bioruptor® Plus sonicator (Diagenode) for 35 cycles (30 sec pulses/30 sec rest) for MDA-

**Fig. 3 | DVL1 is dependent on ETS1 to bind to promoter regions of ETS1 regulated genes. a** RT-qPCR ($n$ = 3 biological replicates (3 technical replicates each), mean ± SD, two-way, ANOVA followed by Dunnett's multiple comparison test was used to test significance) of ETS1 regulated genes ($n$ = 3, log2FC > 2 or < −2, padj < 0.05, statistical analysis and $p$-value calculations were performed using EdgeR,) identified in RNA/ChIPseq integrated analyses, in the presence (DVL1[OE] and DVL1[+/-]) or absence (DVL1[KD] and DVL[-/-]) of DVL1 against a non-targeted control cell line. **b** ChIP and ChIP/re-ChIP ($n$ = 3 biological replicates (3 technical replicates each), mean ± SD, two-way, ANOVA followed by Dunnett's multiple comparison test was used to test significance) in MDA-MB-231, *top*, and HEK293TREx, *bottom*, with various degrees of expression of DVL1. **c** ETS1 regulated gene expression in ETS1 knockdown cells ($n$ = 3 biological replicates (3 technical replicates each), mean ±

SD, two-way ANOVA, followed by Sidak's multiple comparison test was used to test significance. **d** ChIP-qPCR ($n$ = 3 biological replicates (3 technical replicates each), mean ± SD, two-way, ANOVA, followed by Dunnett's multiple comparison test was used to test significance) in MDA-MB-231, *top*, and HEK293TREx-TKO[+/-], *bottom*, control (ETS1[NTC]) and ETS1 knockdown (ETS1[KD]) cells. **e** Illustration showing DVL1 partners with ETS1 in the promoter region of ETS1 regulated genes (*BMP2, CD96, COL1A1, DDX3X, PTEN, SOX4*), *top*; Absence of DVL1 alters DVL1 binding ability to genomic loci but does not affect ETS1 binding, and changes gene expression of ETS1 regulated genes, *bottom left*; knockdown of ETS1 affects DVL1 ability to bind to shared genomic loci and alters gene expression. Created in BioRender. Martinez, D. (2025) https://BioRender.com/ebwkjhc. Source data are provided as the Source Data file.

MB-231 and HEK293TREx DVL TKO. The soluble chromatin fraction (50–150 bp) was quantified and 100 μg of chromatin was immuno-precipitated with anti-DVL1 antibody (D3570, Sigma; 27384-1-AP, Proteintech), anti-HA antibody (cs3724, Cell Signaling), anti-ETS1 antibody (cs14069, Cell Signaling), anti-SIRT1 antibody (cs2493, Cell Signaling) or Rabbit IgG (I5006, Sigma) for 2.5 hrs at 4 °C. Dynabeads Protein A (Invitrogen, 10002D) were added to the chromatin-antibody mixture and incubated with rotation for 2.5 hrs at 4 °C. ChIP samples were washed five times with low salt wash buffer (0.1% SDS, 1% Triton X-100, 2 mM EDTA, 20 mM Tris HCl pH 8.1, and 150 mM NaCl), four times with high salt wash buffer (0.1% SDS, 1% Triton X-100, 2 mM EDTA, 20 mM Tris HCl pH 8.1, and 500 mM NaCl), and once with TE wash buffer (1 mM EDTA and 10 mM Tris HCl pH 8). Immunoprecipitated chromatin samples were reverse crosslinked overnight at 65 °C, followed by RNAse A (Promega) incubation at 37 °C for 2 h, and proteinase K incubation (Promega) at 55 °C for 2 hrs. DNA was eluted using Qiaquick PCR purification kit (Qiagen) and amplified by end-point PCR. Primers used for ChIP-PCR/qPCR are listed in the Source Data File.

## Immunoblots

Antibodies used for immunoblots are: DVL-1 (D3570), DVL-2 (cs3224, Cell Signaling), DVL-3 (cs3218, Sigma), HA (cs3724), ETS1 (cs14069; AB63302, abcam), SIRT1 (cs8469), Lamin A/C (cs4777, Cell Signaling), and GAPDH (cs2778, Cell Signaling). Membranes were incubated with blocking buffer (5% milk/TBST) with primary antibody overnight at 4 °C. This was followed by probing the membranes with HRP-conjugated secondary antibodies for 1 h/room temperature (RT). Lastly, the signals were visualized using ECL reagent (Thermo Scientific) and imaged in the Azure C300 gel imaging system (Azure Biosystems).

## mRNA Expression analysis

Total RNA was isolated using Aurum™ Total RNA Mini Kit (Bio-Rad), and reverse-transcribed using M-MLV Reverse Transcriptase (Promega, M170A) to synthesize the first-strand of complementary DNA (cDNA), using Oligo(dT)20 Primer (Thermo Fisher). Primers were designed for each specific target DNA, and gene expression measured by either endpoint-PCR or real-time RT-qPCR. Primers used for RT-qPCR are listed in the Source Data File.

## End-point PCR and RT-qPCR

End-point PCR amplification was performed using JumpStart RedTaq (Sigma). The Applied Biosystems Veriti 96-well thermal cycler (Applied Biosystems) and Gel DOC EZ imager (Bio-Rad) were used for analyses. Gene expression was quantified by real-time qPCR in CFX96 Normal-Well Real-Time System (BioRad) using PerfeCta SYBR Green FastMix and specific oligonucleotide primers. The reaction mixtures contained 10 μl PerfeCta SYBR Green FastMix, 7.2 μl ddH$_2$O, 2.0 μl template cDNA and 0.5 μl gene-specific 10 μM PCR oligonucleotide primers. The reaction conditions were 95 °C for 30 s, followed by 40 cycles of 95 °C for 5 s and 60 °C for 30 s and Melt Curve (dissociation stage). Relative

gene expression was calculated as delta (Δ Re (the difference between the cycle threshold values, Ct, of the internal control (housekeeping gene, IgG or Input), and Ct of gene of interest) and confirmed by $2^{-\Delta\Delta CT}$ method. Primers used for RT-qPCR and ChIP-qPCR are listed in the Source Data File.

## Subcellular fractionation

MDA-MB-231 and HEK293TREx DVL TKO cells were used for nuclear and cytoplasmic fractionation of DVL1 protein localization. Cell pellets were resuspended in hypotonic buffer (20 mM Tris-HCl, pH 7.4, 10 mM NaCl, 3 mM MgCl$_2$ and Protease Inhibitor cocktail) for 30 min and 25 μL 1% NP-40 was added. Cells were incubated for 30 min before centrifugation at 15,000xg for 10 min, and supernatant (cytoplasmic fraction) collected. Nuclear pellet was washed 4x in PBS and resuspended in either cell extraction buffer (10 mM Tris, pH 7.4, 100 mM NaCl, 1 mM EDTA, 1 mM EGTA, 1 mM NaF, 20 mM Na$_4$P$_2$O$_7$, 2 mM Na$_3$VO$_4$, 1% Triton X-100, 10% glycerol, 0.1% SDS, 0.5% deoxycholate) or Co-IP buffer (25 mM Tris, pH 7.4, 150 mM NaCl, 1 mM EDTA, 1% NP-40, 5% glycerol and Protease inhibitor cocktail) for 45 min and centrifuged at 15,000 x $g$ for 30 min at 4 °C. The supernatant was collected, and both the nuclear and cytoplasmic portion was used for subsequent co-immunoprecipitation.

## Co-Immunoprecipitation

Cells were seeded and grown to confluence in 100 mm dishes. Cells were washed with PBS and lysed in Co-IP lysis buffer (25 mM Tris, pH 7.4, 150 mM NaCl, 1 mM EDTA, 1% NP-40, 5% glycerol and Protease Inhibitor cocktail). This was followed by quantification for equal protein loading using the standard BCA protocol (Thermo Scientific, 23227). Cell extracts were incubated with 2 μg of DVL-1 specific antibody (D3570; Sigma), ETS1 (cs14069, AB63302), SIRT1 (cs8469) specific antibody and species-matched IgG for 2 h/4 °C. Protein A Dynabeads (Life Technologies) were incubated with the antigen-antibody complex for 2 h/4 °C. Beads were washed four times with lysis buffer with gentle agitation for 5 min per wash. 5x sample buffer (Invitrogen) was used for elution of the complex from beads, followed by western blotting along with whole cell extract.

## ChIP-Seq and data analysis

DNA from ChIP was extracted, and at least 10 ng of ChIP DNA was sent to GENEWIZ and used for ChIP-seq library preparation and sequencing. FASTQ files were analyzed using DNASTAR's Lasergene software. Sequenced reads were aligned against the human genome (GRCh38.p14), MACS2[64] and SPP[65] were used for peak calling, and the peaks visualized using integrative genomic viewer (IGV)[66] and ChIPSeeker[67]. Coverage files and putative targets of H3K27ac[68], H3K4me1[68], H3K4me2[69], H3K4me3[70], H3K9ac[71], H3K36me3[69], POL2RA[72] ChIP-seq data from MDA-MB-231 were downloaded from Cistrome Data Browser[73]. De novo DVL1 motif discovery was performed using MEME-ChIP and TOMTOM to identify if those motifs were similar to known consensus sequences using the MEME Suite

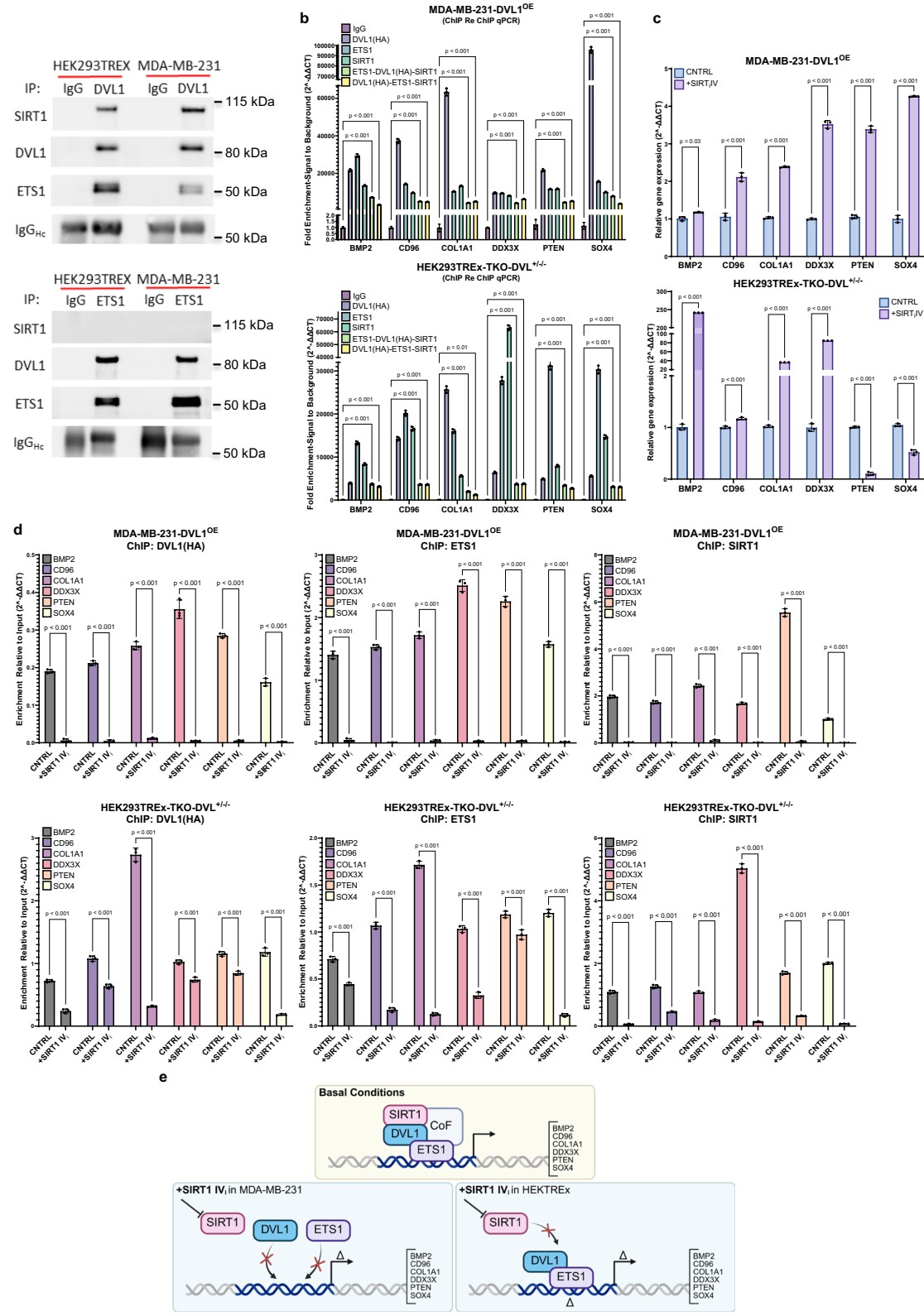

Programs[74]. BETA was used to infer cis-regulatory modules[75]. JASPAR enrichment analysis[76] and Locus Overlap Analysis (LOLA)[77] was used to predict transcription factor binding sites in our dataset.

## RNA-Seq and data analysis

Total RNA was isolated using Aurum™ Total RNA Mini Kit (Bio-Rad), and quality was determined using 2100 Bioanalyzer (Agilent), and quality RNA (RIN > 8) was sent to GENEWIZ for RNA-seq library preparation and sequencing. Differential Gene Expression analysis was done utilizing EdgeR, and its associated packages[78]. Read counts were obtained from GENEWIZ, file annotation with *biomaRt (ENSEMBL)*, and lowly expressing genes (CPM < 1) were filtered prior to further analysis. *Voom* was used as quality control to maintain the false discovery rate below the nominal rate. Normalization for composition bias to eliminate biases between libraries was performed using *EdgeR*. Differential expression and

**Fig. 4 | SIRT1 partners with DVL1 at ETS1 genomic loci. a** Co-Immunoprecipitation for endogenous DVL1 and ETS1, and probe for endogenous DVL1, ETS1 and SIRT1 in the cytoplasmic and nuclear fraction. Representative blots shown (three biological replicates); IgG heavy chain was the loading control for IP. **b** ChIP and ChIP/re-ChIP/re-ChIP ($n$ = 3 biological replicates (3 technical replicates each), mean ± SD, two-way, ANOVA, followed by Dunnett's multiple comparison test was used to test significance) for HA-DVL1, ETS1, and SIRT1 in the same genomic region in MDA-MB-231 and HEK293TREx cells. **c** RT-qPCR identifies significant ($n$ = 3 biological replicates (3 technical replicates each), mean ± SD, two-way, ANOVA followed by Sidak's multiple comparison test was used to test significance) alteration of gene expression with SIRT1 IV inhibitor in MDA-MB-231 and HEK293TREx cell lines. **d** ChIP-qPCR ($n$ = 3 biological replicates (3 technical replicates each), mean ± SD, two-way, ANOVA, followed by Sidak's multiple comparison test was used to test significance) of HA-DVL1, ETS1, and SIRT1 in ETS1-regulated genes in both cell lines. **e** Illustration demonstrating SIRT1 partners with DVL1, and DVL1 partners with both ETS1, SIRT1, and other possible co-factors (CoF) at the same genomic loci in the promoter to regulate gene expression, *top*; inhibition of SIRT1 in MDA-MB-231 prevents binding of DVL1 and ETS1 to genomic loci, and alters gene expression, *bottom right*; inhibition of SIRT1 in HEK293TREx changes DVL1 and ETS1 binding potential, altering gene expression. Created in BioRender. Martinez, D. (2025) https://BioRender.com/ebwkjhc. Source data are provided as the Source Data file.

gene set testing was performed using *limma* and *TREAT* and further visualized with *glimma* and *EnhancedVolcano*.

### Data mining and pathway analysis

Pathway analysis and gene set enrichment analysis was done using R. R packages utilizing BioCarta, GO (gProfiler[79]), KEGG, Reactome (ReactomePA[80]), and GSEA databases were used. Pathway visualization plots were done using Cytoscape ClueGo[81] and Cerebral[82].

### Reporting summary

Further information on research design is available in the Nature Portfolio Reporting Summary linked to this article.

## Data availability

The RNA-seq and ChIP-seq data generated in this study have been deposited in the Gene Expression Omnibus database (https://www.ncbi.nlm.nih.gov/geo/) under accession code GSE249323. The histone data used in this study are available in the Cistrome Data Browser under accession codes: H3K27ac, H3K4me1, H3K4me2, H3K4me3, H3K9ac, H3K36me3, POL2RA. The remaining data are available within the Article, Supplementary Information or Source Data file. Source data are provided in this paper.

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

## Acknowledgements

We would like to acknowledge Vítězslav Bryja for sharing HEK293TREx DVL TKO cell lines generated by his lab. This research was supported by the National Cancer Institute of the National Institutes of Health under award number T32CA19589 to D.M.M., the National Institutes of Health (CA155223) to K.P., Cancer Prevention and Research Institute of Texas (CPRIT), Recruitment of Rising Stars Award (RR140008) to K.P.

## Author contributions

D.M.M. and K.P. designed the study and supervised the project. M.S. generated key reagents, and M.S. and D.M.M. validated them in vitro. M.S., D.M.M. performed the ChIP-seq experiments. D.M.M. and J.V.W. performed the RNA-seq experiments. D.M.M. conducted all bioinformatics analyses. J.V.W. and F.S.M. quantified gene expression. D.M.M., J.V.W., and F.S.M. performed the ChIP experiments, D.M.M., G.P.B., E.J., G.S., and R.L.B. performed the WB and co-IP experiments. D.M.M. and K.P. wrote the manuscript and incorporated revisions and edits from manuscript co-authors at all stages.

## Competing interests

The authors declare no competing interests.
