## [Transparent Peer Review file · Nature Communications]

Dishevelled-1 regulates global transcriptomic changes and associates with ETS1 transcription factor

Corresponding Author: Dr Kevin Pruitt

Version 0:

Reviewer comments:

Reviewer #1

(Remarks to the Author)

This manuscript by Martinez-Marín and colleagues presents some very interesting and provocative data about a potential nuclear role for Dvl1 in mediating transcription via interaction with ETS and SIRT1. While this is a very interesting concept and could be highly cited after publication, there are some key information that should be included to further enhance the manuscript.

Major points.

1. Independent of how the ETS1-Dvl1 interaction was first identified, it is important to know whether this interaction is specific for DVL1 or applies to all three DVL genes in mammals. Did the authors evaluate whether DVL2 and DVL3 can also associate with ETS1?
2. The authors make an important observation that they can identify endogenous DVL1 localizing to promoters via CHIP-seq. However, when it comes time to show an association between Dvl1 and ETS1 they use HA tagged DVL1 to evaluate this. Can the authors show an association between endogenous Dvl1 and Ets1?
3. Is there information on what domain in Dvl1 mediates the direct binding to ETS1?
4. Would Dvl1 knockdown or knockout affect gene expression in a reciprocal manner to those genes whose expression is altered by Dvl1 overexpression?
5. What proportion of Dvl1 Chip-seq associations are dependent on ETS1? In other words, if you knocked down ETS1 would you expect a large majority of the Chip-seq hits to disappear?

Minor Points

6. The authors report that approximately half the transcriptome (~11,000 transcripts) is changed upon overexpression of Dvl1. It is likely that a large percentage of this is not due to Dvl1 associating with nuclear transcriptional targets. In fact, they show that approximately 3,100 genes can be seen with Dvl1 Chip-seq showing association with those promoters. The fact that a lot of these changes are not due to nuclear Dvl1 should be more clearly stated. Ideally, one would express a Dvl1 variant that can't be localized to the nucleus, but given that the key point of this paper is that Dvl1 interacts with Ets1, I don't think that is necessary.
7. Do the authors have any speculation on whether it is monomeric Dvl1 that is associating with nuclear targets or are there oligomers that mediate this?

Reviewer #2

(Remarks to the Author)

In the submitted manuscript, Martinez-Marín and colleagues investigate the activity and function of DVL1 in the nucleus of human cells. Building up on previous findings by the authors and others, this study unravels a genome-wide binding pattern of DVL1, and the ability of DVL1 to influence the expression of a large number of genes. DVL1, likely not possessing its own DNA binding ability, relies on other transcription factors who tether it along the chromatin. Among these ETS1, whose presence is required for the physical association of DVL1 to target loci. Finally, the authors identify a functional cooperation between ETS1-DVL1 and the epigenetic regulator SIRT1, broadening the spectrum of action and interest on DVL1.

The article has the merit of investigating a so far understudied role of DVL proteins, previously mainly considered in their capacity of cytosolic regulators of Wnt signaling.

Moreover, several biochemical analyses (ChIP and re-ChIP) combined with overexpression and mutational studies are persuasive in building a model where DVL1 is a powerful gene regulator that hitchhikes other DNA-binding proteins, among which ETS1.

Future work is needed and desirable to understand what tissues and biological processes are affected by this novel DVL1 function and its partnership with ETS1 and SIRT1.

Below I list a series of comments that, if solved, would in my opinion significantly consolidate the article.

1) What is the status of Wnt signaling activation when nuclear DVL1 regulates target genes? Does nuclear DVL1 deprive the pool of the Wnt signalling-relevant cytosolic one? I encourage the authors to share more about their ideas and explanations on the new nuclear DVL1 function and its relation to Wnt signalling.

2) DVL was found to assemble on the promoters of Wnt targets together with beta-catenin and TCF/LEF (DOI: 10.1083/jcb.200710050). Is perhaps DVL1 a constitutive player of the beta-catenin transcriptional complex?

3) With such a beautiful experiment as DVL1 ChIP-seq it is a shame that all the peaks are hidden and compressed in Extended Figure 2. I recommend the authors to show more of this analysis, including details of the relevant peaks on the loci that are followed up in subsequent figures. Importantly, zooming into the ChIP-seq tracks would allow the authors and readers to understand more about my previous point: are there DVL1 ChIP-seq peaks on Wnt responsive elements within the AXIN2 locus, and other classical Wnt targets?

4) Have the authors measured the amount of DVL1 overexpression? It would be interesting to know whether the achieved abundance falls within a physiological range of DVL1 expression that exists across tissues. This would avoid the (trite, yet relevant) criticism of protein squelching caused in overexpression assays.

5) The authors use TREAT to focus on genes presenting a significant change including those whose mRNA variation is of 10%. The logic for this choice is not clear: while these 10% changers could be of interest, mildly affected mRNAs likely provide small contributions to the DVL1 overexpression phenotype, and I would consider more interesting to focus first on the mRNAs that change more, at least as a first approach, and considering that the gene expression variation induced by DVL1 appears to be consistent.

6) I do not fully understand the need of verifying with ChIP-qPCR the binding of DVL1 on the promoters shown in Figure 1D. If these promoters present ChIP-seq enrichment peaks, it would be advisable to show these peaks before or in parallel to the qPCR validation (which I would not, as a general comment, consider necessary, as ChIP-seq experiments are in my opinion intrinsically more reliable than ChIP-qPCR assays). Moreover, the qPCR analysis will at best validate the presence of DVL1 but will not provide any additional information that DVL1 cooperates with factors whose motif had been found in the vicinity, as it seems to be pointed out by the authors at line 97 "to further validate our in silico analysis" referring to the de novo motif search.

As a general suggestion, I would suggest showing the peaks emerging from the ChIP-seq analysis. Their shape and size and relation to the background is a good (the best?) way to evaluate the overall quality of the experiment thus lending credibility to each individual instance and all subsequent analyses.

7) In the ChIP-qPCR of Figure 1D it is not clear what antibody was used. I assume the anti-DVL1, given the prominent signal on these promoters even in the EV condition where there is no DVL1-HA?

8) I find unclear Extended Figure 2D: what datasets have been used to compare DVL1 binding to these chromatin marks? The legend says: "Factors with high giggle scores are likely to regulate the given genes." Yet what do the authors mean by "factor", and how could one score what "genes" are included in this analysis is cryptic. I suggest the authors to provide a better explanation for this analysis and its implications.

9) Extended Figure 2E is not clear, and it seems that its caption is missing. Does the distance from TSS indicated in the table refers to DVL1, to these core promoter elements, or to when these two are simultaneously present? For instance, that TATA-box is typically found at circa -30 nucleotides is textbook knowledge and it is not clear how this information is relevant here.

10) Extended Figure 3 is interesting, yet simple analyses would improve its readability and what could be concluded from it. For example, as there appears to be roughly equal numbers of unregulated and downregulated genes, this might be an opportunity to test which processes are activated and which repressed by DVL1 by "simply" looking for patten of DVL1 promoter occupancy and activation or repression of the respective gene.

11) At line 126 the authors describe the identification of transcription factors that might cooperate with DVL1. It is not clear to me if this includes the analysis of actual ChIP-seq/CUT&RUN performed on these proteins (ETS or GATA, for example), or if this refers to the identification of ETS and GATA motifs being present within DVL1 peak regions. It would be useful if this part is better described.

12) The pull down presented in Figure 2D is a good experiment, yet it requires a better control, that is the display of a transcription factor and/or cytosolic protein which is expressed at amounts comparable to ETS but it is not precipitated by the

anti-HA-DVL1. Given that the authors have used glycerol 5% in their buffer, one could expect proteins to become more "sticky". An overexpressed DVL1 would further enhance this effect by crowding.
Was the endogenous, non overexpressed DVL1 in MDA-MB-231 cells tested for its interaction with endogenous ETS1?
Also for this alternative experiment a good negative control is advisable.

13) In Figure 2E it surprises me not seeing any value/bar for the IgG. One should expect an amplification here as all ChIP experiments carry over genomic DNA material to some extent.
On the same experiment: measuring the enrichment relative to Input is not particularly useful unless negative control regions are included. One should test if the signal is higher in some regions (e.g., on the BMP2 locus) in respect to other control regions where DVL1 does not (or is not supposed to) bind, and this information can be easily obtained by the authors' ChIP-seq.
In absence of this, it is difficult to interpret the results. For example, on the BMP2 locus, DVL1 ChIP-qPCR gives a value inferior to 1. Is this a case of not binding?
Moreover, the two re-ChIP samples (in light and dark green) display a much higher signal: this should not be theoretically possible as, at least the light green DVL(HA)-ETS1 sample must be a fraction of the previous IP (only DVL1(HA)) efficiency. How this is calculated, and if they are measured different Inputs must be clarified.

Report by Claudio Cantù.

Version 1:

Reviewer comments:

Reviewer #1

(Remarks to the Author)

The revision process has improved an already strong manuscript. This should stimulate significant discussion in the Wnt signaling field as to the mechanisms underlying the function of the Dishevelled proteins.

Reviewer #2

(Remarks to the Author)

My former comment (reported below), that the authors might have uploaded as bedgraph an RNA-seq experiment, rather than ChIP-seq, has been responded.

I still remain skeptical about the nature of the peaks represented, which mostly coincide with exons. It is important, in my opinion, for the authors to show evidence that their result does not derive from a contamination by cDNA libraries.

My suggestions are:

- To add the sequenced track of the Input sample in the main figure
- To add the sequenced track of the IgG negative control in the main figure
- To show other genomic loci where the peaks identified are located in genuine regulatory regions, such as enhancers and promoters
- Provide a global view of the genomic location of peaks (how many in exons, TSS, intergenic), such that any reader or user, if skeptical of the exonic peaks, could computationally extract the different categories.

I also suggest that the authors emphasise their awareness that exonic peaks are unexpected and potentially artifactual (as cDNA contamination could in principle affect any single reagent, including the anti-DVL1 antibody used), or, if they are confident in their controls and tools, that these peaks might underly a yet difficult to interpret relation of DVL1 with the chromatin (e.g., could co-transcriptional splicing be captured by ChIP-seq?).

My former comment - left here for the record:

Looking at that the ChIP-seq peaks, now displayed in Figure 1C and Figure 4 - Extended figure 2C in response to one of my former comments, I wonder if the authors perhaps did upload to IGV the mapped reads of their RNA sequencing experiment instead? The "peaks" are almost invariably in exonic regions with very sharp borders that could be typical of cDNA libraries but not of sonicated ChIP-seq DNA. I encourage the authors to double check this and to render available the bedGraph files of their ChIP-seq experiment such that with no analysis required a reviewer/reader can upload them in IGV and evaluate their quality.

Version 2:

Reviewer comments:

Reviewer #2

(Remarks to the Author)

1) I am still a little worried by noticing that the tracks displayed in figures 1C and extended figure 2C-D show IgG peaks exactly in the same positions as DVL3 ones.

While apparently smaller, these IgG peaks are sharp and characterized by high signal-to-background ratio (they appear being genuine peaks surrounded by low background reads). Presumably, these regions have been identified as DVL3 peaks by peak calling over the IgG tracks. While it is true that ChIP-seq measures local enrichment (e.g., anti-DVL3/IgG), it is of concern if the genome-wide profile of the IgG sample almost exactly matches the experimental track. The concern is exacerbated by the fact that this applies to all tracks/genomic positions chosen for the figures, and by the observation that none of these potential peaks is ornamented by relevant chromatin marks (extended Figure 2C-D).

My suggestion is to find a way to exclude peaks where there is considerable signal in the IgG at the exact same position. One way to do this is to call DVL3 and IgG ChIP-seq peaks each against the input separately, and then consider valid the DVL3 peaks that are not present in the IgG (by subtraction).

One last note: while possibly demanding, I believe that the ultimate test for the authors to make sure about the quality of the dataset is the repetition of the experiment, possibly with a different antibody. I strongly encourage them to do so.

2) While I personally remain skeptic about the exonic peaks, I like how the authors support and document their potential validity. In extended figure 2B, I encourage them to separate them from promoters and intronic and UTR.

Version 3:

Reviewer comments:

Reviewer #2

(Remarks to the Author)

The authors have performed additional tests that remove my previous concerns. In particular, the ChIP-seq experiment is presented and performed with independent reagents and antibodies, guaranteeing a higher standard of reproducibility.

May 30, 2024

Re: NCOMMS-23-38356A

Dear Reviewers,

Please find below our point-by-point response, wherein we have incorporated all of the reviewer recommendations. This has indeed improved the quality of our manuscript.

Below is our point-by-point response (in red) to each of the questions/concerns raised by the reviewers. In addition, the revisions in the new manuscript are also highlighted in red text color.

RESPONSE TO REVIEWERS' COMMENTS

Reviewer #1

1. Independent of how the ETS1-Dvl1 interaction was first identified, it is important to know whether this interaction is specific for DVL1 or applies to all three DVL genes in mammals. Did the authors evaluate whether DVL2 and DVL3 can also associate with ETS1?

Response: We appreciate the inquiry from Reviewer #1 regarding the interaction of ETS1 with other DVL paralogs. We have observed and documented ETS1's association with different DVL paralogs in the revised manuscript, Supp Figure 1B. However, we have not yet elucidated the specific functions of DVL2 or DVL3 in relation to their interaction with ETS1.

2. The authors make an important observation that they can identify endogenous DVL1 localizing to promoters via CHIP-seq. However, when it comes time to show an association between Dvl1 and ETS1 they use HA tagged DVL1 to evaluate this. Can the authors show an association between endogenous Dvl1 and Ets1?

Response: Thank you for your comment. We show interaction between endogenous DVL1 and endogenous ETS1 by IP of DVL1 followed by blot of ETS1 (Fig 4A) and by IP of ETS1, followed by blot of DVL1 (Fig 4B). We have revised Figure 4 to better illustrate the association between endogenous DVL1 and ETS1.

3. Is there information on what domain in Dvl1 mediates the direct binding to ETS1?

Response: Thank you for your inquiry. Currently, it is unclear which specific domain is responsible for the association between DVL1 and ETS1. Identifying this could provide valuable insights, and we are planning to explore this in during the next phase of investigation.

4. Would Dvl1 knockdown or knockout affect gene expression in a reciprocal manner to those genes whose expression is altered by Dvl1 overexpression?

Response: We appreciate the feedback from the reviewer. In response, we have revised Figure 3A to more clearly illustrate the gene expression associated with DVL1 gain-of-function (OE) and loss-of-function (KD).

5. What proportion of Dvl1 Chip-seq associations are dependent on ETS1? In other words, if you knocked down ETS1 would you expect a large majority of the Chip-seq hits to disappear?

Response: This is an excellent and important question we have considered. Given the new reagents needed to properly control this experiment as well as the cost, we would like to address this in a separate follow-up investigation. We were hoping that this report would serve as the primary innovation in identifying the first transcription factor that partners with nuclear DVL1. However, to address this as best we can at the moment, we have expanded the discussion section.

Also, considering this important question, while DVL1 is associated with several ETS1-regulated genes, it does not interact with all 12,000 target genes of ETS1. Given that DVL1 binds to multiple regions within a gene and likely interact with other transcription factors, follow-up investigation could include an analysis of how many hits will disappear with not only ETS1 knockdown but with knockdown of another DVL1-interacting TF that may emerge with continued investigation. Conducting an experiment where ETS1 is knocked down (and another DVL1-interacting TF) is knocked down followed by DVL1 ChIP-Seq/Cut-and-Run could be enlightening, potentially revealing the specific genes that DVL1 and ETS1 jointly regulate.

6. The authors report that approximately half the transcriptome (~11,000 transcripts) is changed upon overexpression of Dvl1. It is likely that a large percentage of this is not due to Dvl1 associating with nuclear transcriptional targets. In fact, they show that approximately 3,100 genes can be seen with Dvl1 Chip-seq showing association with those promoters. The fact that a lot of these changes are not due to nuclear Dvl1 should be more clearly stated. Ideally, one would express a Dvl1 variant that can't be localized to the nucleus, but given that the key point of this paper is that Dvl1 interacts with Ets1, I don't think that is necessary.

Response: This is an excellent observation. We have now detailed the alterations in the transcriptome attributable to DVL1 in our discussion section.

7. Do the authors have any speculation on whether it is monomeric Dvl1 that is associating with nuclear targets or are there oligomers that mediate this?

Response: We appreciate the reviewer's insightful question. Currently, we have not determined whether the DVL is in its monomeric form. To address this, we are planning an experiment in which we will introduce DVL1 that is either FLAG or HA tagged to see the extent of monomeric vs. oligomeric association. This approach will facilitate the investigation and provide clearer insights into the molecular state of DVL during its association with nuclear targets.

Reviewer #2

1. What is the status of Wnt signaling activation when nuclear DVL1 regulates target genes? Does nuclear DVL1 deprive the pool of the Wnt signalling-relevant cytosolic one? I encourage the authors to share more about their ideas and explanations on the new nuclear DVL1 function and its relation to Wnt signalling.

Response: We value the reviewer's inquiry and have expanded the discussion to address how mutants of DVL1 with altered nuclear entry/exit signals could provide deeper insights into the specific role of DVL1 localization in regulating Wnt signaling pathways.

2. DVL was found to assemble on the promoters of Wnt targets together with beta-catenin and TCF/LEF (DOI: 10.1083/jcb.200710050). Is perhaps DVL1 a constitutive player of the beta-catenin transcriptional complex?

Response: Thank you for your inquiry. We have addressed this significant question in our discussion and have also expanded on it by including additional binding motifs for TCF4 in Extended Figure 2E.

3. With such a beautiful experiment as DVL1 ChIP-seq it is a shame that all the peaks are hidden and compressed in Extended Figure 2. I recommend the authors to show more of this analysis, including details of the relevant peaks on the loci that are followed up in subsequent figures. Importantly, zooming into the ChIP-seq tracks would allow the authors and readers to understand more about my previous point: are there DVL1 ChIP-seq peaks on Wnt responsive elements within the AXIN2 locus, and other classical Wnt targets?

Response: Thank you for the valuable feedback and insightful suggestions. Based on your recommendations, we have added extra ChIP-seq tracks to more clearly illustrate the DVL1 peaks on classical Wnt target genes. This enhancement helps to further substantiate our findings and provide a clearer visual representation of the data.

4. Have the authors measured the amount of DVL1 overexpression? It would be interesting to know whether the achieved abundance falls within a physiological range of DVL1 expression that exists across tissues. This would avoid the (trite, yet relevant) criticism of protein squelching caused in overexpression assays.

Response: Thank you for your inquiry. In response, we have included additional blots, Supp Figure 1A, in the supplementary figure to address this important observation.

5. The authors use TREAT to focus on genes presenting a significant change including those whose mRNA variation is of 10%. The logic for this choice is not clear: while these 10% changers could be of interest, mildly affected mRNAs likely provide small contributions to the DVL1 overexpression phenotype, and I would consider more interesting to focus first on the mRNAs that change more, at least as a first approach, and considering that the gene expression variation induced by DVL1 appears to be consistent.

Response: Thank you for your feedback. To improve clarity, we have revised our description of how TREAT is employed to select the genes that are used in downstream analysis and experiments.

6. I do not fully understand the need of verifying with ChIP-qPCR the binding of DVL1 on the promoters shown in Figure 1D. If these promoters present ChIP-seq enrichment peaks, it would be advisable to show these peaks before or in parallel to the qPCR validation (which I would not, as a general comment, consider necessary, as ChIP-seq experiments are in my opinion intrinsically more reliable than ChIP-qPCR assays). Moreover, the qPCR analysis will at best validate the presence of DVL1 but will not provide any additional information that DVL1 cooperates with factors whose motif had been found in the vicinity, as it seems to be pointed out by the authors at line 97 “to further validate our in silico analysis” referring to the de novo motif search. As a general suggestion, I would suggest showing the peaks emerging from the ChIP-seq analysis. Their shape and size and relation to the background is a good (the best?) way to evaluate the overall quality of the experiment thus lending credibility to each individual instance and all subsequent analyses.

Response: Thank you for your valuable feedback. In response, we have incorporated the enrichment peaks into our analysis.

7. In the ChIP-qPCR of Figure 1D it is not clear what antibody was used. I assume the anti-DVL1, given the prominent signal on these promoters even in the EV condition where there is no DVL1-HA?

Response: Thank you for your feedback. We have updated the manuscript to clearly specify the antibodies used in each experiment.

8. I find unclear Extended Figure 2D: what datasets have been used to compare DVL1 binding to these chromatin marks? The legend says: “Factors with high giggle scores are likely to regulate the given genes.” Yet what do the authors mean by “factor”, and how could one score what “genes” are included in this analysis is cryptic. I suggest the authors to provide a better explanation for this analysis and its implications.

Response: Thank you for your inquiry. We updated the text and Extended Figure 2E (now Extended Figure 3C) to include additional clarification to better highlight our findings' significance.

9. Extended Figure 2E is not clear, and it seems that its caption is missing. Does the distance from TSS indicated in the table refers to DVL1, to these core promoter elements, or to when these two are simultaneously present? For instance, that TATA-box is typically found at circa -30 nucleotides is textbook knowledge and it is not clear how this information is relevant here.

Response: Thank you for your feedback. We have updated the caption for Extended Figure 2E and included additional details on the input data used by the computational model to identify DVL1 binding to specific nucleotide sequences within a defined distance from the transcription start site (TSS)

10. Extended Figure 3 is interesting, yet simple analyses would improve its readability and what could be concluded from it. For example, as there appears to be roughly equal numbers of unregulated and downregulated genes, this might be an opportunity to test which processes are activated and which repressed by DVL1 by “simply” looking for pattern of DVL1 promoter occupancy and activation or repression of the respective gene.

Response: Thank you for your suggestions. In response, we have updated Extended Figure 3 to include an additional ranking method for the various processes/pathways identified in our sequencing analysis.

11. At line 126 the authors describe the identification of transcription factors that might cooperate with DVL1. It is not clear to me if this includes the analysis of actual ChIP-seq/CUT&RUN performed on these proteins (ETS or GATA, for example), or if this refers to the identification of ETS and GATA motifs being present within DVL1 peak regions. It would be useful if this part is better described.

Response: Thank you for your feedback. We have provided further clarification on how the LOLA analysis incorporates the DVL1 ChIP-seq data to pinpoint potential transcription factors.

12. The pull down presented in Figure 2D is a good experiment, yet it requires a better control, that is the display of a transcription factor and/or cytosolic protein which is expressed at amounts comparable to ETS but it is not precipitated by the anti-HA-DVL1. Given that the authors have used glycerol 5% in their buffer, one could expect proteins to become more “sticky”. An overexpressed DVL1 would further enhance this effect by crowding. Was the endogenous, non overexpressed DVL1 in MDA-MB-231 cells tested for its interaction with endogenous ETS1? Also for this alternative experiment a good negative control is advisable.

Response: Thank you for your feedback. We have revised the manuscript to clarify the text and images that illustrate the binding of endogenous DVL1 to endogenous ETS1. The figure that addresses this is Figure 4a.

13. In Figure 2E it surprises me not seeing any value/bar for the IgG. One should expect an amplification here as all ChIP experiments carry over genomics DNA material to some extent. On the same experiment: measuring the enrichment relative to Input is not particularly useful unless negative control regions are included. One should test if the signal is higher in some regions (e.g., on the BMP2 locus) in respect to other control regions where DVL1 does not (or is not supposed to) bind, and this information can be easily obtained by the authors' ChIP-seq. In absence of this, it is difficult to interpret the results. For example, on the BMP2 locus, DVL1 ChIP-qPCR gives a value inferior to 1. Is this a case of not binding? Moreover, the two re-ChIP samples (in light and dark green) display a much higher signal: this should not be theoretically possible as, at least the light green DVL(HA)-ETS1 sample must be a fraction of the previous IP (only DVL1(HA)) efficiency. How this is calculated, and if they are measured different Inputs must be clarified.

Response: This is an insightful observation. We have clarified and refined the calculations for the DVL1 ChIP-qPCR.

Sincerely,

Kevin Pruitt, Ph.D.
Associate Professor
Department of Pharmacology
Lineberger Comprehensive Cancer Center
University of North Carolina at Chapel Hill

October 4, 2024

Re: NCOMMS-23-38356A

Please find below our point-by-point response. We thank reviewers for the excellent feedback and recommendations which have been incorporated. This has significantly strengthened our manuscript, and we hope the reviewers now find these changes satisfactory.

Point-by-point Rebuttal:

Reviewer #2: “My former comment (reported below), that the authors might have uploaded as bedgraph an RNA-seq experiment, rather than ChIP-seq, has been responded. I still remain skeptical about the nature of the peaks represented, which mostly coincide with exons. It is important, in my opinion, for the authors to show evidence that their result does not derive from a contamination by cDNA libraries.”

Author response: We thank the reviewer for this inquiry and recommendation. We are confident that the results don't derive from cDNA library contamination for the following reasons: (1) The library prep was performed commercially (Azenta) and the integrity of the sample was confirmed by the vendor; (2) the chromatin DNA submitted for NGS analysis was done separately and was not performed the same time as RNAseq analysis. Additionally, generation of the chromatin for ChIPseq and generation of the RNA for RNAseq were performed by different co-authors, and both sets of samples were further prepped by a vendor and were submitted years apart; (3) RNase A was used as part of the ChIP protocol to further purify DNA for sequencing, and during NGS library prep from DNA submitted for ChIPseq analyses, no reverse transcription takes place; (4) Prior to this current DVL1 ChIPseq, we performed several prior unpublished DVL ChIPseq using both academic and commercial vendors to optimize the protocol.

Reviewer #2: “My suggestions are:

- To add the sequenced track of the Input sample in the main figure
- To add the sequenced track of the IgG negative control in the main figure
- To show other genomic loci where the peaks identified are located in genuine regulatory regions, such as enhancers and promoters”
- Provide a global view of the genomic location of peaks (how many in exons, TSS, intergenic), such that any reader or user, if skeptical of the exonic peaks, could computationally extract the different categories.

Author response: We thank the reviewer for another excellent recommendation. First, we have now added sequenced tracks for Input and IgG in the figures and these data are now also accessible to the public. Second, to provide a global view of the genomic loci where the peaks identified in regulatory regions, we have revised Fig.1C to focus on the promoter region associated with the regulatory regions such as INR, CAAT, TATA for simplicity. Moreover, we have included additional data in the extended figures (Extended Fig.2C and D) showing DVL1 in enhancers, exon, intron, and intergenic regions with the inclusion of IgG, Input, histone, and refseq annotation tracks.

Reviewer # 2: “I also suggest that the authors emphasise their awareness that exonic peaks are unexpected and potentially artifactual (as cDNA contamination could in principle affect any single reagent, including the anti-DVL1 antibody used), or, if they are confident in their controls and tools, that these peaks might underly a yet difficult to interpret relation of DVL1 with the chromatin (e.g., could co-transcriptional splicing be captured by ChIP-seq?).”

Author response: We thank the reviewer for this recommendation which has been incorporated. We have added references to support possible explanations and have also better discussed alternative interpretations of our data. For example, numerous studies such as several shown here and cited in the manuscript (Rambout 2018, Caggiano 2019) show that ChIP has been used to reveal the recruitment of auxiliary splicing components and components of the canonical spliceosome with target intron/exon boundaries as part of co-transcriptional splicing. Second, ChIPseq experiments of transcription factors have detected the assembly of spliceosome components on individual genes. Third, RNA Polymerase II (RNAPII) association with splicing factors at most introns initiate their splicing cotranscriptionally (Naftelberg 2015). Fourth, ChIP-seq data can be used to identify the specific DNA sequences that correspond to increased frequency of factors associated with intron retention events (Agirre 2021, Braunschweig 2013). Hence, because splicing components enriched at intron/exon junctions and RNA binding proteins are preserved on nascent RNA with formaldehyde cross-linking, some peaks may be associated with splice junctions. While beyond the scope of the present manuscript, LC-MS experiments identifying DVL1 binding partners have revealed several factors involved in co-transcriptional splicing bind to DVL1. This unpublished data support the role of DVL1 serving as a scaffold for proteins that regulate co-transcriptional splicing. We plan to submit this follow-up study as a separate independent study.

Reviewer #2: “My former comment - left here for the record: Looking at that the ChIP-seq peaks, now displayed in Figure 1C and Figure 4 - Extended figure 2C in response to one of my former comments, I wonder if the authors perhaps did upload to IGV the mapped reads of their RNA sequencing experiment instead? The “peaks” are almost invariably in exonic regions with very sharp borders that could be typical of cDNA libraries but not of sonicated ChIP-seq DNA. I encourage the authors to double check this and to render available the bedGraph files of their ChIP-seq experiment such that with no analysis required a reviewer/reader can upload them in IGV and evaluate their quality.”

Author Response from former comment: We appreciate the inquiry from the Reviewer. There were no errors in the ChIPseq data files we uploaded as described above; however, we converted the figures into the *.bedgraph format. We have now uploaded ChIP-seq data to GEO and it is now fully accessible to the public

Sincerely,

Kevin Pruitt, Ph.D.
Associate Professor
Department of Pharmacology
Lineberger Comprehensive Cancer Center
University of North Carolina at Chapel Hill

April 16, 2025

Re: NCOMMS-23-38356A

Please find below our point-by-point response. We thank reviewers for excellent recommendations which have been incorporated. We hope the further strengthened manuscript is now fully satisfactory.

Point-by-point Rebuttal:

Reviewer #2: “I am still a little worried by noticing that the tracks displayed in figures 1C and extended figure 2C-D show IgG peaks exactly in the same positions as DVL3 ones. While apparently smaller, these IgG peaks are sharp and characterized by high signal-to-background ratio (they appear being genuine peaks surrounded by low background reads). Presumably, these regions have been identified as DVL3 peaks by peak calling over the IgG tracks. While it is true that ChIP-seq measures local enrichment (e.g., anti-DVL3/IgG), it is of concern if the genome-wide profile of the IgG sample almost exactly matches the experimental track. The concern is exacerbated by the fact that this applies to all tracks/genomic positions chosen for the figures, and by the observation that none of these potential peaks is ornamented by relevant chromatin marks (extended Figure 2C-D). My suggestion is to find a way to exclude peaks where there is considerable signal in the IgG at the exact same position. One way to do this is to call DVL3 and IgG ChIP-seq peaks each against the input separately, and then consider valid the DVL3 peaks that are not present in the IgG (by subtraction).”

Author response: We thank the reviewer for this inquiry and recommendation. We have now repeated the DVL1 ChIPseq analysis using the same DVL1 antibody purchased from Sigma (D3570). This first DVL1 antibody was used for the initial DVL1 ChIPseq. For the initial ChIPseq and the repeat ChIPseq, the library prep was performed commercially (Azenta), and the integrity of the sample was confirmed by the vendor. For our repeat DVL1 ChIPseq repeat, the chromatin DNA was submitted for NGS analysis with no other samples being processed. In the first DVL1 ChIPseq, the chromatin prep was completed by Monica Sharma using D3570 while we were at TTUHSC. In this 2nd analysis, the chromatin prep was completed by Dalia Martinez-Marin, Ph.D. at UNC. The fact that the chromatin analyzed was generated independently by Dr. Sharma and Dr. Martinez-Marin, different manuscript co-authors, further strengthens our finding.

Another difference is that in the repeat DVL1 (D3570) ChIPseq, we used a different Bioruptor at UNC than the first analysis (at TTUHSC). We also ensured that fresh RNase A was used as part of our standard ChIP protocol. These differences across different institutions and different co-authors further ensure the integrity of our findings. These findings reveal that DVL1 associates with multiple loci across the genome of MDA-MB-231 cells. While not included in this manuscript, but which will be part of a separate independent study, we also find a greater extent of steady-state DVL1 nuclear localization relative to the other paralogs (DVL2 or DVL3). This suggests an even more widespread nuclear role of DVL1 in some cancer cell lines. While all three paralogs have both a nuclear localization signal and a nuclear export signaling, little is known about each of their nuclear functions, so our study provides significant new insight into the nuclear role of DVL1.

Reviewer #2: “One last note: while possibly demanding, I believe that the ultimate test for the authors to make sure about the quality of the dataset is the repetition of the experiment, possibly with a different antibody. I strongly encourage them to do so..”

Author response: We thank the reviewer for another excellent recommendation which has now incorporated. We performed a third DVL1 ChIPseq with another antibody from Proteintech (27384-1-AP). In Fig 1C we now show DVL1 ChIPseq using D3570 (Sigma) and 27384-1-AP (Proteintech) with similar binding properties across representative loci. While the first ChIPseq involving D3570 was performed at TTUHSC, and the second ChIPseq involving D3570 and 27384-1-AP was performed at UNC, we see the same trends revealing that DVL1 associates with multiple genomic loci in breast cancer cells. Like with the D3570 datasets, we will deposit the new D3570 and 27384-1-AP datasets in GEO.

Reviewer # 2: “While I personally remain skeptic about the exonic peaks, I like how the authors support and document their potential validity. In extended figure 2B, I encourage them to separate them from promoters and intronic and UTR.”

Author response: We thank the reviewer for the careful analyses, and we welcome the helpful and healthy skepticism which serves as an impetus to make analyses more thorough and robust. We have now revised Fig 1C and extended figures. Over the last three months we have made the analysis more robust, and again, thank the reviewer for these recommendations which have made the entire analysis more rigorous.

Sincerely,

Kevin Pruitt, Ph.D.
Associate Professor
Department of Pharmacology
Lineberger Comprehensive Cancer Center
University of North Carolina at Chapel Hill